# Systemic Antimicrobial Treatment of Chronic Osteomyelitis in Adults: A Narrative Review

**DOI:** 10.3390/antibiotics12060944

**Published:** 2023-05-23

**Authors:** Rok Besal, Peter Adamič, Bojana Beović, Lea Papst

**Affiliations:** 1Department of Infectious Diseases, University Medical Centre Ljubljana, 1000 Ljubljana, Slovenia; 2Faculty of Medicine, University of Ljubljana, 1000 Ljubljana, Slovenia

**Keywords:** chronic osteomyelitis, antibiotic therapy, route of administration, duration of antibiotic therapy, rifampicin, adverse reactions

## Abstract

Chronic osteomyelitis in adults is a complex condition that requires prolonged and intensive antimicrobial therapy, but evidence on optimal selection and duration of antibiotics is limited. A review of PubMed and Ovid Embase databases was conducted to identify systematic reviews, meta-analyses, retrospective and randomised controlled trials (RCTs) on antibiotic treatment outcomes in adults with chronic osteomyelitis. Three main areas of interest were investigated: short-term versus long-term antibiotic therapy, oral versus parenteral antibiotic therapy, and combination antibiotic therapy with rifampicin versus without rifampicin. A total of 36 articles were identified and findings were synthesised using a narrative review approach. The available literature suffers from limitations, including a lack of high-quality studies, inconsistent definitions, and varying inclusion/exclusion criteria among studies. Most studies are open-labelled and lack blinding. Limited high-quality evidence exists that oral therapy is non-inferior to parenteral therapy and that shorter antibiotic duration might be appropriate in low-risk patients. Studies on the impact of rifampicin are inconclusive. Further well-designed studies are needed to provide more robust evidence in these areas.

## 1. Introduction

Chronic osteomyelitis is a complex and challenging condition characterised by persistent infection of bone tissue that often requires prolonged and intensive antimicrobial therapy [1,2]. It is caused by microbial invasion of bone tissue by local wound contamination or by hematogenous spread, often with biofilm formation. Microbial invasion is followed by an immune response to the infection and subsequent damage to bone trabeculae and matrix, as well as vascular channels [2,3]. Confirmation of the diagnosis requires several diagnostic procedures, including clinical and laboratory findings, microbiological and histological investigations, and imaging studies such as CT and MRI [2].

The treatment of osteomyelitis requires an interdisciplinary approach, which involves patient evaluation, antibiotic therapy, and debridement of infected tissue and surgical resection of necrotic bone or prosthetic material. The presence of necrotic bone is a hallmark of chronic osteomyelitis and requires surgical debridement before successful antimicrobial treatment. If surgery is not possible, long-term oral antimicrobial suppression of the infection may be necessary [4]. The choice of antibiotic treatment regimen is a critical decision in the management of chronic osteomyelitis, with several different approaches available [1,2,5,6]. In this narrative review, we aim to provide a comprehensive overview of the existing evidence on three important aspects of systemic antimicrobial treatment for chronic osteomyelitis in adults.

The first point of interest is optimal duration of antibiotic therapy. Chronic osteomyelitis is typically treated with at least 6 weeks of (parenteral) antibiotic therapy [2]. However, shorter-term therapy, ranging from 4 to 6 weeks or less, has been proposed as an alternative, with no definitive consensus [7,8,9]. Secondly, the role of oral antibiotic therapy remains a topic of debate. Oral antibiotics or short-term parenteral therapy followed by oral antibiotics are a potential management option for chronic osteomyelitis and could provide an alternative to parenteral therapy alone [10,11]. Finally, rifampicin is often considered as an adjunctive treatment option for chronic osteomyelitis, especially when caused by *Staphylococcus aureus* [12]. However, clear evidence is lacking.

Through this narrative review, we aim to first synthesise the existing data and knowledge available on these three major points of interest and provide a short and concise review of currently available articles, which can serve as a base and convenient overview for future researchers studying this topic and designing studies. Secondly, we aim to critically evaluate the available evidence on these points of systemic antimicrobial treatment for chronic osteomyelitis and determine if specific conclusions can be drawn, given the strength of evidence presented in the studies. Our goal is for this review to be an asset to clinicians treating patients with chronic osteomyelitis in the absence of clear international guidelines.

## 2. Results

### 2.1. Short-Term versus Long-Term Antibiotic Therapy

16 articles were identified in relation to the first research question, including 1 meta-analysis, 3 randomised controlled trials (RCTs) (Table 1) and 12 retrospective studies.

A study of 351 adult patients by Bernard et al. found no difference between 6-week and 12-week antibiotic regimens for treating hematogenous vertebral osteomyelitis. Patients with vertebral implants, recurrence of osteomyelitis, absence of definitive microbiological identification, life expectancy less than 1 year and fungal, brucellar or mycobacterial infection were excluded. However, non-inferiority of a 6-week regimen was not true for some subgroups: patients older than 75 years, in infections due to microorganisms other than *Staphylococcus aureus*, when immunosuppression or diabetes mellitus were present, in the presence of infective endocarditis, neurological signs, abscesses, in post-surgical osteomyelitis and when rifampicin and fluoroquinolone were prescribed. Patients aged 75 years or older and those with *S. aureus* infection had a higher risk of treatment failure, independently of the duration of antibiotic treatment. No significant difference in the proportion of patients with treatment failure was found between patients who received intravenous treatment for more than a week and patients who received intravenous treatment for less than a week. Adverse reaction rates were similar between the two groups (29% vs. 29% respectively) [13].

A study by Tone et al. compared 6-weeks versus 12-weeks of antibiotic treatment of diabetic foot osteomyelitis in 40 adult patients. Results showed no significant difference in rates of remission between 6-weeks and 12-weeks of antibiotic treatment (60% vs. 70% respectively). The study included only patients, who were treated nonsurgically and did not have severe arterial stenosis. No parameters were associated with the patient’s outcome. In this study, gastrointestinal adverse events were reported in significantly more patients receiving 12 weeks of antibiotic treatment than in those receiving only 6 weeks (45% vs. 15% respectively). No patients had *Clostridium difficile* associated diarrhoea [14].

A study by Gariani et al. on the other hand compared 93 adult patients with diabetic foot osteomyelitis, who underwent surgical debridement. A 3-week antibiotic regimen was compared with a 6-week regimen. Patients with total clinical amputation of all infected tissue, patients with destruction of the bone beyond the cortical level and those with concomitant infection at a different site requiring more than 21 days of treatment were excluded. The study found that a relatively short course of post-debridement antibiotic therapy was non-inferior in terms of achieving remission to the duration of 6 weeks (84% vs. 73%) and the number of adverse effects was similar in both groups (38.6% vs. 32.7%) [15].

One meta-analysis by Huang et al. found no significant difference in the rate of treatment failure in patients with osteomyelitis treated with short-course regimens compared with long-course regimens. This meta-analysis consisted of 5 open-label randomised controlled trials and 10 observational studies, 7 studies were focused on vertebral osteomyelitis, 6 on acute osteomyelitis in childhood, 1 study on chronic osteomyelitis and 1 study on diabetic foot osteomyelitis [6]. The only study on chronic osteomyelitis by Rod-Fleury et al. was a retrospective study [16].

The meta-analysis included studies which enrolled patients with osteomyelitis who underwent an antibiotic regimen regardless of the presence of another intervention and compared outcomes between a longer and a shorter course of antibiotics. Short course was defined as a period, shorter than the recommended 4–6 weeks of treatment. Studies, which included osteomyelitis caused by *Brucella*, *Mycobacterium* were excluded. Non-inferiority was shown in both RCTs and observational studies. Regarding vertebral osteomyelitis, there was a higher rate of treatment failure in patients treated with short-course antibiotic therapy, but there was large heterogeneity among studies. One of the reasons for heterogeneity was the proportion of *S. aureus* infection, which related to a higher risk of treatment failure in patients, treated with short-course antibiotics. A higher risk of treatment failure was also present in patients with epidural abscesses, osteomyelitis at additional sites and patients with diabetes mellitus. A study on diabetic foot osteomyelitis showed no difference in outcome between short- and long- course antibiotic therapy. The same goes for chronic osteomyelitis if adequate sequestrectomy is performed. The authors’ conclusion was that a shorter course of antibiotics may be appropriate in low-risk patients. In patients with *S. aureus* infection, vertebral osteomyelitis, or other comorbidities considered as high-risk, at least 6 weeks of antibiotic therapy appears to be necessary [6].

12 retrospective observational studies were identified, which are summarised briefly. Of the 12 retrospective studies 5 were already included in the meta-analysis by Huang et al. [6] (Roblot et al., Park et al., Chang et al., Locke et al., Rod-Fleury et al.) [9,16,17,18,19].

Adverse drug reactions in patients with vertebral osteomyelitis were studied by Kim et al. in a retrospective study, which found that adverse infections were more often present when antibiotics duration was longer [20]. A study by Schindler et al. also concluded that the total duration of antibiotic therapy and parenteral administration were associated significantly with adverse effects in patients receiving long-term antibiotic therapy for osteoarticular infection, while the occurrence of *C. difficile* diarrhoea was low (3.8%) [21].

Three additional retrospective studies of vertebral osteomyelitis were identified. Findings from Li et al., a retrospective study of 109 cases, suggest that short-term antibiotic therapy is non-inferior in low-risk patients, while recurrence of infection was more often present among the high-risk patients receiving short-term therapy (56.2% vs. 22.2% respectively), like conclusions from the meta-analysis by Huang et al. [22]. Two further studies were descriptions of case series (Priest et al., Graham et al.), which suggested that a treatment of at least 6–8 weeks was associated with better clinical outcomes [23,24].

The only retrospective study on chronic osteomyelitis by Rod-Fleury et al. included 49 episodes of chronic osteomyelitis in adult patients. Neither longer duration of therapy (>6 weeks) nor length of initial IV duration before PO switch had a significant impact on remission during a median follow-up of 7.2 years [16].

### 2.2. Oral Antibiotic Therapy or Short-Term Parenteral Therapy (<2 Weeks) Followed by Oral Antibiotic Therapy vs. Parenteral Antibiotic Therapy

13 articles were identified in relation to the second research question, including 6 RCTs (Table 2), 3 meta-analyses or systematic reviews and 4 retrospective studies.

The study by Li et al. (OVIVA) enrolled 1054 patients and compared intravenous and oral antibiotic therapy for the first 6 weeks of treatment for bone or joint infection, including patients with infection of osteosynthetic material. Adult patients who, in the attending physician’s opinion, would ordinarily receive 6 weeks of parenteral therapy, were randomly assigned to receive either intravenous or oral antibiotics to complete the first 6 weeks of antibiotic treatment. Excluded were patients with *S. aureus* bacteraemia, bacterial endocarditis, concomitant infection which would require prolonged parenteral antibiotic therapy, infection for which there are no suitable antibiotic choices to permit randomisation, mild osteomyelitis, septic shock, evidence of being unlikely to comply, and mycobacterial, fungal, parasitic, or viral aetiology of the infection. Oral antibiotic therapy was found to be non-inferior to intravenous antibiotic therapy. Treatment failure at 1 year occurred in 14.6% in the intravenous group and 13.2% in the oral group. The median hospital stay was significantly longer in the intravenous group and had more complications associated with the intravenous catheter than the oral group (9.4% vs. 1.0% respectively). No subgroup analyses showed an outcome advantage of either intravenous or oral therapy. There was no significant difference in the incidence of *C. difficile* associated diarrhoea or the percentage of participants reporting at least one serious adverse event [25].

Azamgarhi et al. performed an observational cohort study of 328 patients 12 months before and after implementing the findings of the OVIVA study into clinical practice. Treatment failure occurred in 13.6% of patients in the pre-implementation and 18.6% in the post-implementation group. The difference was not statistically significant. Adverse drug reaction related hospital readmission rates were similar (2.1 and 2.2% respectively) [26].

A study by Gentry et al. compared oral treatment with ofloxacin and parenteral treatment of chronic osteomyelitis. This study included 33 patients with chronic osteomyelitis, patients with osteosynthetic material were excluded. 19 patients received oral ofloxacin and 14 patients received parenteral cephalosporins. Parenteral antibiotics used were either cefazolin or ceftazidime, with the parenteral regimen chosen according to the in vitro susceptibilities of the pathogens isolated from bone cultures. Long term follow up was conducted 18 months after completion of antibiotic therapy. Ofloxacin was statistically non-inferior to parenteral antibiotics. 7 subjects suffered from adverse reactions in the oral group and 4 in the parenteral group [27].

Another study by Gentry et al. compared oral ciprofloxacin and parenteral therapy with broad-spectrum cephalosporins or a nafcillin-aminoglycoside combination. 31 patients were treated with ciprofloxacin and 28 patients treated with intravenous antibiotics. All infections had been surgically debrided prior to enrolment. Patients received intravenous antibiotics between 4 and 6 weeks, whereas therapy with ciprofloxacin lasted 6 weeks and was prolonged if the signs of osteomyelitis persisted. There were no statistically significant differences in clinical efficacy between ciprofloxacin and parenteral antibiotics. Treatment of five of the six polymicrobial osteomyelitis cases involving *Pseudomonas aeruginosa* failed, whereas none of the 9 single-pathogen treatments of osteomyelitis failed. There was one adverse reaction in the ciprofloxacin group and four adverse reactions in the intravenous group [28].

Gomis et al. performed a study comparing oral ofloxacin to parenteral imipenem-cilastatin in the treatment of osteomyelitis. 32 patients with osteomyelitis, caused by microorganisms susceptible to ofloxacin and imipenem-cilastatin were included. 69% of patients in the ofloxacin group and 50% in the imipenem-cilastatin group had a complete cessation of signs of inflammation, the difference was not statistically significant. Inclusion and exclusion criteria were unclear [29].

A study by Euba et al. evaluated long term results in 50 consecutive patients with *S. aureus* non-axial osteomyelitis after extensive surgical debridement, with or without associated osteosynthetic material, treated with either 8 weeks oral co-trimoxazole or 6 weeks of parenteral oxacillin, followed by 2 weeks of oral oxacillin. Patients with polymicrobial infections, resistant isolates and prosthetic joint infections were excluded. During the 10-year follow up, three relapses (11%) occurred in the oral group and two (10%) in the parenteral group. Retention of osteosynthetic material was associated with higher relapse rate [30].

A meta-analysis by Wald-Dickler et al. found no difference in the clinical efficacy of oral versus intravenous (IV)-only antimicrobial therapy. 8 prospective RCTs were included in their meta-analysis, totalling 1321 adult patients. Four trials included patients with osteosynthetic material, most trials excluded axial osteomyelitis, one trial included 39 patients with surgery for vertebral osteomyelitis/discitis. Six trials compared a fluoroquinolone with or without an oral rifampicin to various IV regimens, one study compared oral trimethoprim-sulfamethoxazole (TMP-SMX) plus rifampicin to IV cloxacillin and one study compared standard IV regimens to varied oral regimens. Trials had varied exclusion criteria, including retained osteosynthetic material, bacteremia or sepsis, comorbidities, or secondary sites of infection. Severe adverse drug reactions were similar between oral and IV regimens in 5 trials, in 2 trials adverse reactions were more common in the IV group. Most patients, who were given oral antibiotics, were treated with fluoroquinolones or TMP-SMX, with or without adjunctive rifampicin. Most adverse events from oral therapy in published trials were due to fluoroquinolones [10].

Conterno et al. performed a systematic Cochrane review and meta-analysis of studies regarding antibiotic treatment of chronic osteomyelitis. Included were 8 small randomised controlled trials, totalling 282 participants. Four studies were included in further meta-analysis. They found no statistically significant difference in remission rates at 12 months between groups receiving oral vs. parenteral antibiotic treatment (77% vs. 81% respectively). Moderate to severe adverse events were observed in 6.1 and 9.5% of participants respectively with no statistically significant difference. They concluded that in cases of susceptible bacteria, the route of administration may not affect disease remission, however high-quality data is lacking, and the majority of included studies were over 20 years old. Studies also lacked details on type and adequacy of surgical intervention, affecting comparability between trials. Almost all included trials were judged to be at moderate to high risk of bias due to trial design and reporting [1].

Passerini et al. performed a small retrospective cohort study along with a systematic review of oral versus parenteral antibiotic treatment of native bacterial vertebral osteomyelitis. They identified 13 studies, including 1 RCT and 12 observational studies. The RCT by Li et al. has already been presented in this narrative review. The meta-analysis of seven observational comparative studies did not have enough evidence to show a higher efficacy of either group. Within the observational studies the authors determined high risk of bias due to differences in demographics, comorbidities, and other confounders between the patients in the two arms [11].

Four additional retrospective studies were identified, which have not yet been included in meta-analyses and are summarised briefly.

Babouee et al. performed a retrospective study on 69 consecutive patients with primary spinal osteomyelitis. Excluded were patients with endocarditis, immunodeficiency, orthopaedic implants, and surgical site infections. Early switch to oral antibiotics was performed in 74% of cases after a median parenteral therapy of 18 days. Treatment failure at 1 year was 3% [31].

In Stumphauzer et al., a retrospective study of 257 adult patients with osteomyelitis, the authors found that there was no significant difference in rates of clinical treatment failure between the group receiving parenteral antibiotic (35%, *n* = 257) and the group which was switched to oral antibiotics after 2 weeks of IV treatment (25%, *n* = 20) [32].

Lang et al. performed a retrospective study on 40 patients with osteomyelitis, 33 with associated implant infections and 7 with native osteomyelitis. 17 were treated with all parenteral therapy and 23 received initial oral therapy or were switched to oral therapy after 14 days. Excluded were patients with prior treatment failure and no identified pathogens. No treatment failures occurred in the oral antibiotics group, whereas 35% of the parenteral group experienced treatment failure. The results are not statistically significant, as most patients in the parenteral group had prosthetic joint or other hardware infection and the oral group had fewer [33].

Melis et al. performed a retrospective observational study of 142 cases of osteomyelitis in a single-centre, 24.6% were associated with implant infection. Full recovery was achieved in 72 cases, of which 36 patients received parenteral antibiotic therapy and 43 oral antibiotic therapy, with no significant difference between the groups. Polymicrobial infection and antibiotic treatment shorter than 6 weeks were independently associated with increased chance of treatment failure [34].

### 2.3. Combination Antibiotic Therapy with Rifampicin vs. Antibiotic Therapy without Addition of Rifampicin

7 articles were identified in relation to the third research question, including 3 meta-analyses and 4 retrospective studies. No additional prospective randomised trials were identified.

Ma et al. performed meta-analysis of five randomised controlled trials and two retrospective cohort studies on humans on the role of adjunctive treatment of *S. aureus* bacteremia with deep infections. The meta-analysis also included in-vitro investigations and animal model studies. They concluded that the addition of rifampicin to standard therapy did not decrease incidences of death, rates of bacteriologic failure or relapse [35]. Out of 5 RCT in humans, only three included patients with osteomyelitis. Two were published by Van der Auwera et al. between 1983 and 1985 [36,37] and included fewer than 30 patients. Outcomes for patients with osteomyelitis were not analysed separately. The third RTC by Ruotsalainen et al. showed lower mortality rates for patients with deep infection who received adjunctive rifampicin in addition to standard therapy (17% versus 38%). However, patients who did not receive rifampicin were significantly older and more often had underlying chronic disease or hospital-acquired infection. Patients who did not receive rifampicin had fewer deep infections and cases of endocarditis [38].

A meta-analysis was performed by Perlroth et al. which identified five randomised controlled trials and two retrospective cohort studies in humans on the role of adjunctive treatment of *S. aureus* bacteremia with deep infections. Four studies included adult patients with osteomyelitis without implant infection [39]. Two studies were by Auwera et al. as previously described [36,37]. An additional 2 studies by Norden et al. between 1983 and 1986, which included 28 and 18 patients respectively were described. The studies included patients with surgical implants and excluded patients with significant renal or hepatic disease and those using anticoagulants. The addition of rifampicin did not prove superior to monotherapy with nafcillin. The meta-analysis concluded that human trials have occasionally demonstrated beneficial results of adjunctive rifampicin, but studies were underpowered [40,41].

A systematic review and meta-analysis was performed by Stengel et al. to analyse therapy for bone and joint infections. Three studies comparing rifampicin combinations to monotherapy were identified within the systematic review [42]. A study by Norden et al. concluded that the addition of rifampicin did not prove superior to monotherapy with nafcillin [41]. A study by Zimmerli et al. found monotherapy with ciprofloxacin inferior to combination therapy with rifampicin in patients with chronic staphylococcal bone infections, but these findings were specific to orthopaedic implant infections [12].

Three additional retrospective studies not included in the meta-analyses were identified. Two were by Wilson et al. studying patients with diabetic foot osteomyelitis treated with or without the addition of rifampicin to standard therapy. Combined endpoints were mortality or amputation within 2 years of follow-up. Patients receiving rifampicin had lower rates of mortality or amputation in both studies (26.9% vs. 37.2% *n* = 6174 in study 1 and 29% vs. 38% *n* = 10,736 in study 2) [43,44]. However, both studies were limited by a small rifampicin arm (130 and 151 patients, respectively) and other confounders. Patients in the rifampicin arm were younger, with fewer comorbidities, had more infectious disease consultations performed during treatment, and more often had *S. aureus* identified in cultures [43,44].

Another study by Wang et al. retrospectively analysed recurrence rates of osteomyelitis in 902 patients after surgical debridement and placement of antibiotic cement, followed by different antibiotic regimes. Patients were treated either with 2 weeks of parenteral antibiotics (IV group), 2 weeks of parenteral, followed by 4 weeks of oral antibiotics (oral group) or 2 weeks of parenteral, followed by 4 weeks of oral antibiotics with the addition of rifampicin (rifampicin group). Recurrence rates of osteomyelitis in the oral and rifampicin group were similar (10.1% and 10.5%, respectively). Adverse effects were more frequent in the rifampicin group than the oral group. Elevated alanine transaminase (ALT) levels were present in 27.4% vs. 18.0% respectively and proteinuria was present in 9.3% vs. 4.5% respectively [45]. The study also found that patients who received parenteral antibiotics for only 2 weeks had fewer adverse effects (elevated ALT 15.1%, proteinuria 3.2%) and a higher recurrence rate (17.9%) [45].

## 3. Materials and Methods

Research questions were identified using a PICO strategy (Patient/Problem, Intervention, Comparison and Outcome). We included adult patients with osteomyelitis, unrelated to implant infection, in all three PICO questions. The outcomes studied were mortality, relapse rate, and adverse effects. The first intervention studied was short-term antibiotic therapy of less than 6 weeks, compared to long-term therapy. The second intervention studied was oral antibiotic therapy or short-term parenteral therapy (less than 2 weeks) followed by oral antibiotic therapy, compared to parenteral therapy alone. The third intervention of interest was combination antibiotic therapy with rifampicin for treating staphylococcal infections, compared to antibiotic therapy without the addition of rifampicin.

A comprehensive literature search was conducted using electronic databases, including PubMed and Ovid Embase, to identify relevant articles. The search strategy utilised a combination of keywords and Medical Subject Heading (MeSH) terms related to chronic osteomyelitis and systemic antimicrobial treatment options (Appendix A). No language or publication date restrictions were applied.

After retrieving the initial search results, duplicate articles were removed. Titles and abstracts of the remaining articles were screened independently by two reviewers (Adamič, Besal) based on predetermined inclusion and exclusion criteria using the Rayyan platform. Included were all articles studying any of the three points of interest in adults with osteomyelitis. Exclusion criteria for studies were studies focusing specifically on orthopaedic implant-related infection, osteomyelitis in paediatric populations, in vitro and animal model studies, studies of infections, other than osteomyelitis, and studies of osteomyelitis in adults which did not include any of the key points of interest.

Full-text articles of the potentially relevant studies identified during the title and abstract screening were retrieved and further reviewed by the same two reviewers to determine their eligibility for inclusion in the narrative review. Studies that provided relevant information on short-term versus long-term antibiotic therapy, oral antibiotic therapy or short-term parenteral therapy followed by oral antibiotic therapy versus parenteral antibiotic therapy, or combination antibiotic therapy with rifampicin versus antibiotic therapy without the addition of rifampicin were included in the review. Any discrepancies were resolved through discussion and consensus.

Data from the included studies were extracted and summarised in a standardised format. Information on study design, sample size, patient characteristics, intervention details, outcomes assessed, and study findings related to the three major points of interest were extracted.

Given the heterogeneity in study designs, populations, interventions, and outcomes among the included studies, a narrative review approach was employed to synthesise the findings. Key findings and trends from the included studies were summarised and discussed in relation to the research questions. The limitations and strengths of the included studies and the overall evidence were also critically appraised.

A total of 6721 citations were identified from the literature search. After screening the titles and abstracts, 6544 articles were excluded based on the predetermined inclusion and exclusion criteria. The full-text review of the remaining 177 articles resulted in the inclusion of 36 studies in the narrative review (Figure 1). Mendeley software was used as a reference manager.

In the analysis and results reported, specific focus and highlights were put on higher-quality randomized controlled trials (RCTs) and meta-analyses, which carry a greater significance in terms of the strength of evidence. Retrospective observational studies were mentioned and briefly summarized at the end of each segment for context and to provide a full overview of the subject matter. A focus was put on RCTs and meta-analyses when critically evaluating and discussing data.

## 4. Discussion

### 4.1. Short-Term versus Long-Term Antibiotic Therapy

The optimal duration of antibiotic therapy for chronic osteomyelitis remains unclear [5,8]. Since 1970 a duration of antibiotic treatment of at least 4 weeks has been recommended by Waldvogel et al. [46]. Short-term antibiotic therapy, ranging from 4 to 6 weeks, has been proposed as an alternative to long-term therapy, which may extend beyond 6 weeks or even several months [7,8,9]. The balance between achieving effective bacterial eradication and minimising the risks of antibiotic-related adverse effects, emergence of antibiotic resistance, and increased healthcare costs is a key consideration in this decision-making process [6].

In all the identified RCTs, as well as meta-analyses in this review, conclusions indicated non-inferiority of short term versus long term therapy for osteomyelitis. However, the quality of evidence for this is low. There is inconsistency in the definitions of short-term and long-term antimicrobial therapy for chronic osteomyelitis, which contributes to the heterogeneity of the results and hinders the ability to draw firm conclusions. Studies also varied in the inclusion of cases where surgical intervention was performed, as well as description of surgical interventions performed. Some studies excluded higher-risk patients, including patients with secondary sites of osteomyelitis or other infection [13,15], patients with certain comorbidities [13,14], and cases where pathogens were not identified [13], which does not reflect real-world clinical settings. Thirdly, studies also showed that non-inferiority of short-term therapy was not present in certain subgroups of patients. These included patients with vertebral osteomyelitis [35], other sites of infection, presence of *S. aureus* infection, patients with comorbidities like diabetes [13,35] and older patients [13]. Evidence exists that adverse effects were more common in long-term antibiotic therapy [14,20,21]. In conclusion, there exists limited evidence showing that shorter-term therapy may be sufficient for low-risk patients, while in high-risk patients more than 6 weeks antibiotic therapy appears to be necessary. Further prospective placebo-controlled studies in patient populations, reflecting real-world clinical settings, are warranted to determine optimal duration of antibiotic therapy.

### 4.2. Oral Antibiotic Therapy or Short-Term Parenteral Therapy (<2 Weeks) Followed by Oral Antibiotic Therapy vs. Parenteral Antibiotic Therapy

There has been growing interest in exploring the role of oral antibiotic therapy or short-term parenteral therapy followed by oral antibiotics as a potential outpatient management option for chronic osteomyelitis [10,11]. Due to vascular impairment, it is a therapeutic challenge to deliver pharmaceutical agents to the site of osteitis [47,48]. Lew et al. in his study suggests parenteral therapy as the approach of choice, cautioning against the risks and complications associated with intravenous catheters [2]. An oral approach may offer advantages in terms of reduced hospitalisation duration, improved patient convenience, and potentially lower healthcare costs [25,26,49], with no statistically significant cure rates between oral and parenteral therapy [49]. However, the efficacy, safety, and optimal selection of patients for this approach compared to traditional parenteral antibiotic therapy require further investigation [1,11]. Local antibiotic delivery systems present an interesting alternative in osteomyelitis treatment. Future developments in sustained-release antibiotic delivery systems hold promise for reducing costs and improving drug efficacy and patient compliance [47]. However, in this review, we do not focus on evaluating local antibiotic delivery systems.

In our review we have found that limited high-quality studies exist on the impact of route of administration on disease remission in chronic osteomyelitis. In the literature studied, there is evidence that supports non-inferiority of oral to parenteral administration. In a large, multi-centre, open-labelled RCT by Li et. al. (OVIVA) [25] there was no significant difference in rates of treatment failure for both implant-related and non-implant-related bone and joint infections. Adverse reactions were less common in patients receiving oral therapy [25]. This was the only study, identified in this review, that provided statistically significant data in a large patient population, to support the use of oral antibiotic therapy. The study was not, however, double blind or placebo controlled. In other identified RCTs non-inferiority of oral therapy was also demonstrated, but these results were not statistically significant. These studies were limited by small sample sizes, difference of type and duration of therapy between the oral and parenteral arms and various exclusion criteria, which limits their applicability [27,28,29,30,32,33,34]. Our findings suggest oral antibiotic therapy may be an acceptable alternative to parenteral therapy.

### 4.3. Combination Antibiotic Therapy with Rifampicin vs. Antibiotic Therapy without Addition of Rifampicin

Combination antibiotic therapy with rifampicin has been proposed as an adjunctive treatment option for chronic osteomyelitis due to its potent activity against biofilm-forming bacteria, which are known to be a key factor in the pathogenesis of chronic osteomyelitis [12]. The addition of rifampicin to the antibiotic regimen may improve the clinical outcomes and reduce the risk of treatment failure [12]. However, the optimal duration and selection of patients for this approach remain uncertain, and the potential for rifampicin-associated adverse effects, including drug interactions and development of rifampicin resistance, needs to be carefully considered [39].

Despite the clinical interest in the use of combination antibiotic therapy with rifampicin in chronic osteomyelitis, there is a scarcity of RCTs examining its effectiveness. While some studies have evaluated the use of combination antibiotic therapy with rifampicin in implant-associated bacterial infections, there is a paucity of studies specifically focusing on native bacterial osteomyelitis. Implant-associated infection and native bacterial osteomyelitis have differences in terms of pathophysiology, microbial profile, and treatment approach. Therefore, the findings from implant-associated infection studies may not be directly applicable to native bacterial osteomyelitis, and more research is needed. In our research, some RCTs showed improved treatment outcomes with the addition of rifampicin [12,38,43,44] with non-statistically significant results, while others observed no significant difference [35,40,41,45]. Common limitations of studies included: investigation of *S. aureus* bacteremia without separate analysis of patients with osteomyelitis [36,37], small size, exclusion of patients with comorbidities [40,41], unbalanced arms in terms of size [43,44] or different patient comorbidities in study arms [38,43,44]. In one study adverse effects were more common in patients receiving rifampicin [45]. Given this data, no firm conclusions can be drawn from available studies regarding the role of rifampicin and more research is needed in the future.

### 4.4. Limitations of the Available Literature

In conclusion, chronic osteomyelitis is a complex condition that requires long-term antimicrobial therapy. However, the available literature on this topic suffers from several limitations, which were true for all our investigations. One major limitation is the lack of high-quality studies, particularly randomised controlled trials (RCTs), which are considered the gold standard for evaluating the efficacy of medical interventions. Many available RCTs suffer from small sample sizes and large heterogeneity in the included patient populations.

Most studies in the literature are open-labelled. This can introduce bias and impact the reliability of the findings, as there may be subjective judgments or expectations that influence the outcomes. Blinding studies are often challenging to implement in antimicrobial treatment studies due to the nature of the interventions. This lack of blinding may affect the internal validity of the studies and limit the strength of the conclusions that can be drawn.

The inclusion and exclusion criteria used in the studies also vary widely, which can impact the generalizability of the findings. Some studies excluded high-risk patients, such as those with comorbidities or complicated cases of chronic osteomyelitis, which may not reflect the real-world clinical practice. This selection bias may limit the external validity of the studies and affect the applicability of the findings to broader patient populations.

Another limitation of the existing literature is that most studies are relatively old, with some studies dating back more than 30 years. Medical practices and standards of care have evolved since then, including advances in surgical techniques, imaging modalities, and antimicrobial therapies. The age of the studies may limit their applicability to current clinical practice and may not reflect the most up-to-date evidence. The findings from older studies may also be influenced by factors such as changes in microbial resistance patterns, which may have an impact on the effectiveness of antimicrobial treatment.

Chronic osteomyelitis often requires a combination of medical and surgical interventions for optimal management. However, many studies in the literature lack detailed information on the surgical interventions that were performed, which can affect the comparability of the findings between trials. The type, timing, and adequacy of surgical intervention can significantly impact the outcomes of antimicrobial treatment, and the lack of standardised reporting of surgical interventions in the studies may limit the ability to draw clear conclusions.

## 5. Conclusions

The available evidence from RCTs and meta-analyses seems to indicate non-inferiority of short-term versus long-term antimicrobial therapy for osteomyelitis. Evidence is however limited and suggests that shorter-term therapy may only be sufficient for low-risk patients, whereas high-risk patients require at least 6 weeks of antibiotic therapy. Adverse effects were found to be more common with long-term therapy.

Evidence from the literature suggests that oral administration is non-inferior to parenteral administration. Li et al.’s RCT (OVIVA) showed no significant difference in treatment failure rates between the two routes of administration, with adverse reactions being less common in patients receiving oral therapy. Although other RCTs have also demonstrated the non-inferiority of oral therapy, these results were not statistically significant.

While some studies have evaluated the use of combination antibiotic therapy with rifampicin in implant-associated bacterial infections, there is a paucity of studies specifically focusing on native bacterial osteomyelitis. While some RCTs showed improved treatment outcomes with the addition of rifampicin, others observed no significant difference. No firm conclusions can be drawn from available studies, and more research is needed.

The available literature on chronic osteomyelitis suffers from several limitations that should be considered when evaluating the efficacy of antimicrobial therapy. A major limitation is the lack of high-quality and recent studies, particularly randomised controlled trials, with small sample sizes being a common issue, as well as open-labelled design, variability in inclusion and exclusion criteria and type of surgical intervention. These limitations underscore the need for further high-quality studies and highlight the need for a multidisciplinary approach to its management.

## Figures and Tables

**Figure 1 antibiotics-12-00944-f001:**
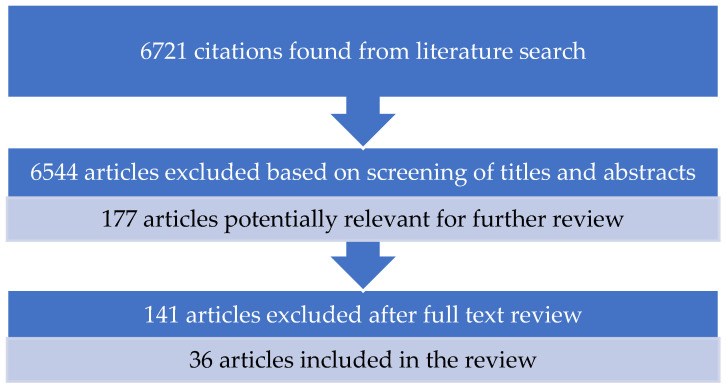
Flow diagram of the selection process.

**Table 1 antibiotics-12-00944-t001:** Randomised controlled trials: Oral antibiotic therapy or short-term parenteral therapy (<2 weeks) followed by oral antibiotic therapy vs. parenteral antibiotic therapy.

Article	Year	Osteomyelitis Type	Sample Size	Antibiotic Duration	Exclusion Criteria	Results
Bernard et al. [13]	2015	Vertebral osteomyelitis	351	6 vs. 12 weeks	Life expectancy of less than 1 year; pregnancy or breastfeeding; presence of a vertebral implant; recurrence of spondylodiscitis; presence of fungal, brucellar, or mycobacterial infection; absence of microbiological identification	Remission occurred in 160 (90·9%) of 176 patients in the 6-week group and in 159 (90·9%) of 175 in the 12-week group.
Tone et al. [14]	2014	Diabetic foot osteomyelitis	40	6 vs. 12 weeks	Significant stenosis or arterial occlusions	Remission occurred in 12 (60%) cases of the patients treated for 6 weeks and in 14 (70%) cases of the patients treated for 12 weeks (*p* = 0.50)
Gariani et al. [15]	2021	Diabetic foot osteomyelitis after surgical debridement	93	3 vs. 6 weeks	Diabetic foot osteomyelitis associated with an implant; effective antibiotic therapy < 96 h previously; amputation of all infected tissue; complete destruction of bone beyond cortical level; remote infection requiring more than 21 days of another antibiotic	Remission occurred in 37 (84%) of the patients in the 3-week arm compared to 36 (73%) in the 6-week arm (*p* = 0.21)

**Table 2 antibiotics-12-00944-t002:** Randomised controlled trials: Oral antibiotic therapy or short-term parenteral therapy (<2 weeks) followed by oral antibiotic therapy vs. parenteral antibiotic therapy.

Article	Year	Osteomyelitis Type	Sample Size	Investigation	Exclusion Criteria	Results
Li et al. [25]	2019	Bone and joint infection, including infection of osteosynthetic material	1054	Standard oral vs. standard IV	*S. aureus* bacteraemia; endocarditis, concomitant infection requiring prolonged antibiotics; mild osteomyelitis; no suitable antibiotic choices to permit randomisation; septic shock; unlikely to comply with trial; mycobacterial, fungal, parasitic, or viral infection	Non-inferiority of oral therapy to parenteral therapy. Treatment failure at 1 year 13.2% vs. 14.6% respectively
Azamgarhi et al. [26]	2021	Bone and joint infection	328	Standard oral vs. standard IV	Not available	OVIVA trial findings can be safely implemented into clinical practice
Gentry et al. [27]	1991	Non-prosthesis osteomyelitis	33	Ofloxacin vs. cephalosporin	Multiple sites of infection, retained prosthetic material, bacteremia	Ofloxacin statistically non- inferior to IV antibiotics
Gentry et al. [28]	1990	Osteomyelitis after surgical debridement	59	Ciprofloxacin vs. beta-lactam + aminoglycoside	Septicemia, MRSA	Ciprofloxacin statistically non-inferior to IV antibiotics
Gomis et al. [29]	1985	Osteomyelitis including prosthesis infection	32	Ofloxacin vs. imipenem	Not available	Ofloxacin non-inferior to IV imipenem-cilastatin
Euba et al. [30]	2009	Non-axial *Staphylococcus aureus* osteomyelitis	50	TMP-SMX + rifampicin vs. cloxacillin	Prosthetic joint infection, polymicrobial	8 weeks of co-trimoxazole non inferior to or 6 weeks of IV + 2 weeks of PO oxacillin

Legend: IV—intravenous, MRSA—Methicillin resistant *Staphylococcus aureus*, TMP-SMX—trimethoprim-sulfamethoxazole.

## Data Availability

No new data were created or analysed in this study. Data sharing is not applicable to this article.

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
