# Peer review of "Systemic Antimicrobial Treatment of Chronic Osteomyelitis in Adults: A Narrative Review"

_antibiotics, 2023, doi:10.3390/antibiotics12060944_

Round 1
Reviewer 1 Report
The article is logical and innovative, but there are some flaws that need to be revised.
The manuscript is entitled “Systemic antimicrobial treatment of chronic osteomyelitis in
adults: a narrative review”In this study,A review of PubMed and Ovid Embase databases was conducted to identify systematic reviews, meta-analyses, retrospective and randomised controlled trials (RCTs) on antibiotic treatment outcomes in adults with chronic osteomyelitis. Three main areas of interest were investigated: short-term versus long-term antibiotic therapy , oral versus parenteral antibiotic therapy , and combination antibiotic therapy with rifampicin versus without rifampicin. A total of 36 articles were identified and findings were synthesised using a narrative review approach. The available literature suffers from limitations, including a lack of high-quality studies, inconsistent definitions, and varying inclusion/exclusion criteria among studies. Most studies are open-labelled and lack blinding. Limited high-quality evidence exists that oral therapy is non-inferior to parenteral therapy and that shorter antibiotic duration might be appropriate in low-risk patients. However, we think there are still some issues in this manuscript need to be corrected by the author before publication.
1.The purpose and meaning of this review are not clearly expressed.
2.Figure1 is left aligned in the paper, so it is reasonable to center it.
3. In the results, there are only results and a few conclusions in each case, lacking in-depth discussion
The article is smooth overall, without many grammatical errors.
Author Response
Thank you very much for your valuable comment and directions in improving the article. We have updated the following, marking it in red for easy access:
- The purpose and meaning of the review were further clarified at the end of the introduction. We further expanded on the specific inclusion and exclusion criteria for articles during initial and subsequent screenings. The introduction was shortened to provide a general overview of the topic. A clearer description of the design of result reporting was added (with a focus on RCTs and meta-analyses) in the materials and methods portion.
- Figures were centered.
- Appropriate sections of the introduction were integrated into the discussion to provide a better and more in-depth overview.
- An overview of the manuscript was performed by the authors for grammatical and spelling errors. A final proofread and corrections were performed by an independent professional with a degree in the English language and experience in proofreading and corrections.
Reviewer 2 Report
The authors have compared and reviewed a large range of studies conducted with osteomyelitis. They found that oral therapies were not significantly lower infectiveness vs IV treatment which is a common misconception. They additionally did not find any significant difference between longer term treatment studies vs shorter term studies. These comparisons are useful and good guide that can help guide future experimental and clinical design of upcoming studies to ensure the most useful data is generated in clinical trials.
The authors have reviewed published studies regarding treatment of osteomyelitis and its outcomes. They reviewed papers as far back as 1983 and as recently as 2022. Over six thousand papers were browsed and narrowed down to 177 before being further reduced to 36 for in depth review and analysis.
They asked the research questions:
Is oral treatment less effective than IV antibiotics?
Is longer treatment better than short term treatment?
In both cases, the authors did not identify a clear trend and there was no significant difference in the outcomes for oral treatment vs IV antibiotics or longer vs short term treatment.
This review can used to help plan and design future clinical studies to ensure the most useful information is learned.
Specific comments:
· Please go over the manuscript for spelling and grammatical errors.
· Please update the references/citations:
o Please review “Staphylococcal Osteomyelitis: Disease Progression, Treatment Challenges, and Future Direction” from 2018. https://doi.org/10.1128/CMR.00084-17
§ This does not have the same research questions as this review intends but can be used to illustrate the state of the field so it should be cited in the background and introduction.
o Local antibiotic delivery systems for the treatment of osteomyelitis – A review, from 2009 https://doi.org/10.1016/j.msec.2009.07.014
§ This does not have the same research questions as this review intends but can be used to illustrate the state of the field so it should be cited in the background and introduction.
Author Response
Thank you very much for your valuable comment and directions in improving the article. We have updated the following, marking it in red for easy access:
- An overview of the manuscript was performed by the authors for grammatical and spelling errors. A final proofread and corrections were performed by an independent professional with a degree in the English language and experience in proofreading and corrections.
- References/citations were updated (with the suggested articles). The introduction was shortened to provide a general overview of the topic. Citations were added for background and context and the appropriate sections of the introduction were integrated into the discussion to provide a better overview. We further expanded on the specific inclusion and exclusion criteria for articles during initial and subsequent screenings. A clearer description of the design of result reporting was added (with a focus on RCTs and meta-analyses) in the materials and methods portion.